# Beyond All-or-Nothing: Anisotropic Evidential Regularization for Robust Medical Out-of-Distribution Detection

## Abstract

Evidential Deep Learning (EDL) employs KL divergence regularization to constrain evidence on non-target classes for out-of-distribution (OOD) detection. However, recent work Re-EDL challenges this design, arguing that KL regularization is nonessential and advocating its complete removal. Through systematic investigation, we identify an all-or-nothing dilemma: neither full constraint nor no constraint provides universal robustness across datasets, and coefficient tuning fails to escape this binary framework. To address this fundamental limitation, we propose Leaky-KL regularization that achieves anisotropic adaptation through selective class-wise leakage—allowing evidence to leak for top-$k$ predicted classes while constraining remaining classes. This mechanism enables different classes to receive adaptive regularization strengths determined by their evidence patterns across mini-batches, achieving directional adjustment in gradient space beyond simple magnitude scaling. Experiments across three medical datasets demonstrate that Leaky-KL consistently outperforms existing EDL variants and surpasses optimal dataset-specific coefficient tuning, validating the fundamental advantages of anisotropic over isotropic regularization.

## 1. Introduction

The deployment of artificial intelligence in clinical diagnosis demands robust mechanisms to handle unexpected inputs, as prediction errors in safety-critical medical applications can directly impact patient treatments (Gutbrod et al., 2025; Hong et al., 2024). While deep learning models have demonstrated impressive performance on in-distribution (ID) data,

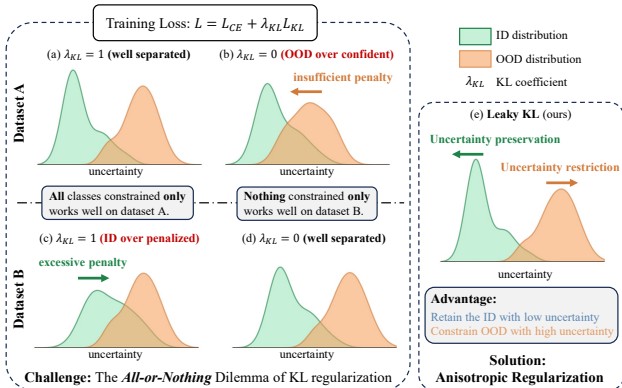

*Figure 1.* **Beyond the existing all-or-nothing dilemma, our Leaky KL achieves dual improvement.** While all constrained excessively penalize ID and nothing constrained insufficiently penalize OOD, Leaky KL can simultaneously retain the uncertainty of ID and constrain OOD with high uncertainty.

they often produce overconfident predictions when encountering out-of-distribution (OOD) samples that deviate from the training distribution (Hendrycks & Gimpel, 2016). In medical imaging, such OOD samples arise from diverse scenarios including rare diseases absent from training data, anatomical variations, and low-quality images with acquisition artifacts (Roy et al., 2022; Schott et al., 2025; Yang et al., 2025). When undetected, these OOD samples significantly undermine clinical utility through misdiagnosis or inappropriate clinical decisions. Therefore, developing reliable OOD detection methods is essential for trustworthy computer-aided diagnosis.

To enable reliable OOD detection, recent work has turned to uncertainty quantification methods that explicitly model prediction confidence. Among these approaches, Evidential Deep Learning (EDL) (Sensoy et al., 2018) has emerged as a promising framework that uniquely combines class prediction and uncertainty quantification within a single forward pass. Grounded in subjective logic theory (Jøsang, 2016), EDL parameterizes a Dirichlet distribution over the class probability simplex, where the concentration parameters, formulated as evidence, naturally capture epistemic uncertainty. To ensure appropriate uncertainty expression, EDL employs a KL divergence regularization term that constrains

[1]Anonymous Institution, Anonymous City, Anonymous Region, Anonymous Country. Correspondence to: Anonymous Author <anon.email@domain.com>.

Preliminary work. Under review by the International Conference on Machine Learning (ICML). Do not distribute.

evidence for non-target classes toward a uniform prior, encouraging higher uncertainty on unfamiliar inputs.

However, the role of KL regularization in EDL has become increasingly controversial. While full constraint ($\lambda_{KL}$=1.0) in Classic EDL has been widely adopted, recent work Re-EDL (Chen et al., 2025) challenges its necessity, advocating complete removal of KL regularization ($\lambda_{KL}$=0.0) as a simple and broadly applicable solution, though acknowledging that tuning $\lambda_{KL}\in(0, 1)$ could achieve better performance. This presents a fundamental question: *is there a universal setting that is suitable for all datasets?*

Through systematic investigation across diverse datasets, we identify an **all-or-nothing dilemma**: neither full constraint ($\lambda_{KL}$=1) nor no constraint ($\lambda_{KL}$=0) can provide stable reliability across diverse scenarios. As schematically illustrated in Figure 1, while the full KL constraint achieves well-separated distributions on Dataset A, it over-penalizes plausible non-target classes in ambiguous ID samples, leading to excessive uncertainty (Fu et al., 2025). Conversely, removing constraint leads to insufficient penalty on outliers, degrading the OOD performance, but it restores proper separation on Dataset B. Moreover, our experiments in coefficient finetuning reveals that the fundamental limitation lies in the **isotropic** nature of classic KL regularization: it treat all non-target classes identically regardless of sample-specific evidence patterns. Despite of achieving better performances, coefficient tuning merely searches for the optimal operating point along the spectrum of "all constrained", without escaping the binary all-or-nothing framework itself.

To address this dilemma, we propose a key insight: the core question is not *how much* to constrain, but *what* to constrain. Based on this insight, we introduce Leaky-KL regularization to achieve **anisotropic regularization**, where different classes receive different effective constraint strengths determined by their collective evidence patterns. For a single input, Leaky-KL selectively allows evidence to "leak" for the top-$k$ predicted classes while maintaining full constraint on remaining classes. Then with the aggregation of minibatch, different effective coefficient are assigned to different classes to achieve class-adaptive constraint. As illustrated in Figure 1(e), this mechanism achieves simultaneous uncertainty preservation for ID samples and uncertainty restriction for OOD samples. For ID samples, leaking evidence for plausible classes preserves low uncertainty; for OOD samples, constraining non-top classes prevents spurious evidence accumulation. By combining these complementary mechanisms, Leaky-KL breaks the isotropic assumption inherent in coefficient tuning and achieves consistent performance across diverse scenarios, opening a new dimension in the solution space beyond magnitude scaling. Our contributions are summarized as follows:

- We identify the all-or-nothing dilemma and reveal that coefficient tuning, through isotropic scaling, fails to resolve this fundamental limitation.

- We propose Leaky-KL to achieve anisotropic regularization through selective leakage, transcending the all-or-nothing dilemma of existing methods.

- We validate that Leaky-KL consistently outperforms existing EDL variants across diverse medical imaging domains with varying characteristics, demonstrating its fundamental advantages.

**Relation to Recent Work**   Our work addresses the all-or-nothing dilemma between classic EDL (Sensoy et al., 2018) and Re-EDL (Chen et al., 2025), proposing anisotropic regularization as a principled solution. We compare against recent EDL variants including improved regularization strategies (Pandey & Yu, 2023; Deng et al., 2023; Chen et al., 2024; 2025) and medical adaptations (Fu et al., 2023; 2025). Comprehensive related work is in Appendix A.

## 2. Preliminaries & Problem Formulation

### 2.1. Evidential Loss Function

Unlike traditional neural networks that produce only point predictions, Evidential Deep Learning (EDL) (Sensoy et al., 2018) models classification outcomes as multinomial distributions through evidence theory and enables simultaneous prediction and uncertainty quantification by parameterizing the conjugate prior Dirichlet distribution. Given an input sample $\mathbf{x}$ for a $K$-class classification task, a neural network $f(\mathbf{x}; \theta)$ produces logits that are transformed into non-negative evidence through an activation function $g(\cdot)$:

$$\mathbf{e} = g(f(\mathbf{x}; \theta)) = [e_1, e_2, \ldots, e_K], \tag{1}$$

where each $e_i$ represents the amount of evidence supporting class $i$. This evidence parameterizes a Dirichlet distribution over class probabilities $\mathbf{p} = [p_1, p_2, \ldots, p_K]$:

$$\mathbf{p} \sim \text{Dir}(\mathbf{p} \mid \boldsymbol{\alpha}), \quad \text{where} \quad \boldsymbol{\alpha} = \mathbf{e} + \mathbf{1}. \tag{2}$$

The concentration parameters $\boldsymbol{\alpha}$ determine the shape of the Dirichlet distribution, with $S = \sum_{i=1}^{K} \alpha_i$ representing the total strength. The model's prediction $\bar{\mathbf{p}}$ and uncertainty $u$ are then derived as the expected value and vacuity of this distribution:

$$\bar{\mathbf{p}} = \frac{\boldsymbol{\alpha}}{S} = \frac{\mathbf{e} + \mathbf{1}}{S}, \tag{3}$$

$$u = \frac{K}{S}. \tag{4}$$

To optimize the model, EDL employs a cross-entropy loss on the expectation of the Dirichlet distribution rather than

directly on point estimates like classic networks. The evidential cross entropy loss is formulated as:

$$\mathcal{L}_{\text{CE}} = \int \left[ \sum_{i=1}^{K} -y_i \log(p_i) \right] \text{Dir}(\mathbf{p}|\boldsymbol{\alpha}) d\mathbf{p}$$

$$= \sum_{i=1}^{K} y_i(\psi(S) - \psi(\alpha_i)), \qquad (5)$$

where $\mathbf{y} = [y_1, y_2, ..., y_K]$ is the one-hot encoded ground truth label and $\psi(\cdot)$ is the digamma function. This loss encourages the model to concentrate evidence on the correct class while naturally expressing uncertainty through the Dirichlet parameterization.

Ideally, an evidential neural network should exhibit distinct behaviors across different scenarios. When a sample clearly belongs to a specific class, the model should output high evidence only for that class. When a sample exhibits ambiguity between classes, the model should assign relatively high evidence to multiple plausible classes. For out-of-distribution samples, the model should output uniformly low evidence across all classes, leading to a near-uniform Dirichlet distribution that expresses high uncertainty.

## 2.2. The all-or-nothing dilemma in KL Regularization

In classic EDL, a regularization term $\mathcal{L}_{\text{KL}}$ is introduced to constrain evidence on non-target categories through KL divergence minimization.

$$\tilde{\boldsymbol{\alpha}} = \mathbf{y} + (1 - \mathbf{y}) \odot \boldsymbol{\alpha}, \qquad (6)$$

$$\mathcal{L}_{\text{KL}} = \text{KL}(\text{Dir}(\mathbf{p} \mid \tilde{\boldsymbol{\alpha}}), \text{Dir}(\mathbf{p} \mid \mathbf{1})). \qquad (7)$$

In the formula above, $\tilde{\alpha}$ masks out the target class, exempting it from regularization. By penalizing deviations from a uniform prior, the KL term pushes the concentration parameters of non-target classes toward 1, thereby increasing model uncertainty and discouraging confident predictions on unfamiliar inputs. While recent work Re-EDL (Chen et al., 2025) advocates for complete removal of this regularization ($\lambda_{\text{KL}}=0$), our systematic investigation shows a more nuanced picture: neither full constraint ($\lambda_{\text{KL}}=1$) nor complete removal provides universal robustness across diverse scenarios.

**The all-or-nothing dilemma.** As illustrated in Figure 2, we observe sharply contrasting behaviors between the two extremes. On the Endoscopy dataset, removing KL constraint ($\lambda_{\text{KL}}=0$) substantially improves both AUROC and FPR@95. Conversely, on the Histopathology dataset, the same removal leads to severe degradation. The root cause lies in dataset-dependent separability characteristics. In Endoscopy, there are significant differences among the categories (samples are shown in Appendix B.1), thus full KL

constraint over-penalizes evidence on plausible non-target classes in ambiguous ID samples, degrading OOD detection. However, the samples in Histopathology shows subtle inter-class variations, and removing constraint leads to multi-class evidence accumulation, producing deceptively low uncertainty on OOD samples. Neither extreme can simultaneously handle both high-separability and low-separability scenarios, revealing the fundamental limitation of uniform regularization strategies.

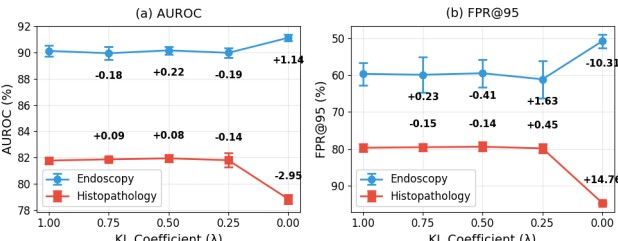

*Figure 2.* **The all-or-nothing dilemma in coefficient finetuning.** Neither extremes provides universal robustness and coefficient tuning cannot escape the binary dilemma.

**The plateau effect in coefficient tuning.** Beyond the dilemma between extremes, Figure 2 reveals that coefficient tuning cannot resolve this fundamental limitation: adjusting $\lambda_{\text{KL}}$ within $(0, 1]$ produces minimal performance changes, with substantial differences appearing only at $\lambda_{\text{KL}}=0$. This **plateau effect** indicates that intermediate coefficients merely find compromise operating points: on high-separability data, any non-zero coefficient over-penalizes confident predictions; on low-separability data, any coefficient below 1.0 under-constrains evidence accumulation. And the coefficient tuning, despite adjusting the magnitude of constraint, remains fundamentally trapped in a binary framework: it uniformly scales constraint across all non-target classes, unable to capture the diverse patterns exhibited by different datasets.

This analysis motivates a fundamental question: since "all constrained" over-penalizes ID samples while "nothing constrained" under-penalizes OOD samples, can we design a method that simultaneously preserves low uncertainty for ID samples while enforcing high uncertainty for OOD samples?

## 3. Leaky-KL Regularization

### 3.1. Method Formulation

The critical question is not *how much* to constrain, but rather *what* to constrain. Unlike scaling the whole regularization with coefficient tuning, we introduce a selective leakage strategy to exempt certain class from constraint based on the sample specific evidence patterns.

Concretely, during training, we first identify the top-$k$ classes with highest evidence values to construct a mask $\mathbf{m}_k$ of "plausible classes":

$$\mathbf{m}_k = \mathbb{I}_{\text{top-}k}(\mathbf{e}), \tag{8}$$

where $\mathbb{I}_{\text{top-}k}(\cdot)$ is an indicator function that returns a binary mask selecting the $k$ classes with maximum evidence values. These top-$k$ classes often captures the sample's inherent ambiguity—although the sample belongs to the target class, it also exhibits high similarity to other classes. Since $\boldsymbol{\alpha}=\mathbf{e}+1$, the top-$k$ operation is directly utilized on $\boldsymbol{\alpha}$ in the following.

Additionally, to prevent incorrect constraint on the target class during early training when predictions may be unreliable, the final leaked masks are defined as the union of $\mathbf{m}_k$ and the ground-truth label:

$$\tilde{\mathbf{m}}_k = \mathbf{m}_k \cup \mathbf{y}. \tag{9}$$

Substituting $\tilde{\mathbf{m}}_k$ for $\mathbf{y}$ in Equation 6, the Leaky concentration parameters can be yielded as:

$$\tilde{\boldsymbol{\alpha}}_{\text{Leaky}} = \boldsymbol{\alpha} \odot (1 - \tilde{\mathbf{m}}_k) + \tilde{\mathbf{m}}_k. \tag{10}$$

With the top-$k$ plausible classes exempted from KL constraint, Leaky-KL achieves sample-adaptive regularization. For samples where certain non-target classes receive high evidence due to genuine ambiguity or visual similarity, exempting these classes preserves legitimate evidence for similar classes and addresses the over-penalty issue. Simultaneously, for the remaining classes with low evidence, maintaining strong regularization suppresses unbounded evidence accumulation and prevents the under-penalty problem.

The Leaky-KL regularization loss is then:

$$\mathcal{L}_{\text{Leaky-KL}} = \text{KL}(\text{Dir}(\mathbf{p} \mid \tilde{\boldsymbol{\alpha}}_{\text{Leaky}}), \text{Dir}(\mathbf{p} \mid \mathbf{1})). \tag{11}$$

The complete training objective combines the evidential classification loss with Leaky-KL regularization:

$$\mathcal{L} = \mathcal{L}_{\text{CE}}(\boldsymbol{\alpha}, \mathbf{y}) + \mathcal{L}_{\text{Leaky-KL}}. \tag{12}$$

Notably, our Leaky-KL formulation **unifies** existing EDL variants as special cases through the leak parameter $k$, regarding the all-or-nothing dilemma as two endpoints of a continuous regularization spectrum. This unification demonstrates that our method is not an ad-hoc modification, but rather a principled framework that generalizes prior approaches:

- When $k=0$, no classes leak beyond the target, recovering standard EDL with full constraint on all non-target classes ($\lambda_{\text{KL}}=1$ extreme).

- When $k=K$, where $K$ is the number of ID classes, all classes leak (combined with the target mask), effectively removing KL regularization and recovering Re-EDL's formulation ($\lambda_{\text{KL}}=0$ extreme).

### 3.2. From Isotropic Scaling to Anisotropic Adaptation

Having established the operational mechanism of Leaky-KL regularization, gradient analysis demonstrates why this approach fundamentally transcends the limitations of coefficient tuning. The key distinction lies in the nature of regularization: coefficient tuning applies a global scalar $\lambda_{\text{KL}}$ uniformly across all samples and non-target classes, a form of regularization termed *isotropic*. In contrast, Leaky-KL generates class-dependent effective regularization strengths that dynamically adapt to mini-batch evidence patterns, a mechanism termed *anisotropic* regularization.

**Sample-Agnostic Constraint in Coefficient Tuning.** For standard KL regularization, consider the gradient with respect to the concentration parameters $\boldsymbol{\alpha}$ of non-target classes. Through the chain rule, we obtain:

$$\frac{\partial \lambda_{\text{KL}} \mathcal{L}_{\text{KL}}}{\partial \boldsymbol{\alpha}} = \lambda_{\text{KL}} \cdot \frac{\partial \mathcal{L}_{\text{KL}}}{\partial \tilde{\boldsymbol{\alpha}}} \cdot \frac{\partial \tilde{\boldsymbol{\alpha}}}{\partial \boldsymbol{\alpha}} = \lambda_{\text{KL}} \cdot \mathbf{g}(\tilde{\boldsymbol{\alpha}}), \tag{13}$$

where $\mathbf{g}(\tilde{\boldsymbol{\alpha}})$ represents the KL gradient structure with respect to $\tilde{\boldsymbol{\alpha}}$ (defined in Eq. 6), and $\partial \tilde{\boldsymbol{\alpha}}/\partial \boldsymbol{\alpha} = \mathbf{I}$ for non-target classes. This formulation exhibits sample-agnostic behavior: all samples sharing the same label receive identical gradient scaling $\lambda_{\text{KL}}$, independent of their individual evidence patterns. Adjusting $\lambda_{\text{KL}}$ uniformly scales all non-target classes by the same factor, while the relative constraint pattern across classes remains invariant.

**Sample-Specific Constraint in Leaky-KL.** In contrast, Leaky-KL introduces sample-specific constraint through the top-$k$ masking mechanism. For non-target classes, the gradient becomes:

$$\frac{\partial \mathcal{L}_{\text{Leaky-KL}}}{\partial \boldsymbol{\alpha}} = \frac{\partial \mathcal{L}_{\text{Leaky-KL}}}{\partial \tilde{\boldsymbol{\alpha}}_{\text{Leaky}}} \cdot \frac{\partial \tilde{\boldsymbol{\alpha}}_{\text{Leaky}}}{\partial \boldsymbol{\alpha}}$$
$$= \mathbb{I}[\notin \text{top-}k] \cdot \mathbf{g}(\tilde{\boldsymbol{\alpha}}_{\text{Leaky}}), \tag{14}$$

where $\mathbb{I}[\notin \text{top-}k]$ is a binary indicator vector and varies across samples based on their evidence rankings. Unlike coefficient tuning's uniform scaling, Leaky-KL applies class-selective scaling: classes within the top-$k$ receive zero gradient, while those outside receive the full scaling $\lambda_{\text{KL}}$. Different samples with the same label may have different top-$k$ sets, constituting sample-specific regularization.

**Mini-Batch Aggregation: From Isotropic to Anisotropic.** The fundamental distinction between isotropic and anisotropic regularization emerges when examining gradient

aggregation across mini-batches. For mathematical simplicity, we assume the mini-batch contains $K$ classes with $N$ samples per class, totaling $KN$ samples. We analyze the batch-averaged gradient for a specific non-target class $i$ by averaging over all samples where class $i$ is non-target.

For standard KL regularization, substituting Eq. 13 into the batch average yields:

$$\frac{1}{KN} \sum_{n=1}^{KN} \frac{\partial \lambda_{\text{KL}} \mathcal{L}_{\text{KL}}^{(n)}}{\partial \alpha_i^{(n)}} = \frac{K-1}{K} \cdot \lambda_{\text{KL}} \cdot \bar{g}_i, \qquad (15)$$

where $\bar{g}_i$ represents the average gradient structure for class $i$ across the non-target samples. Crucially, the coefficient $(K - 1/K)\lambda_{\text{KL}}$ is identical for all classes—it depends only on the global scalar $\lambda_{\text{KL}}$ and the batch composition $K$, disregarding the class-specific characteristics. This constitutes **isotropic regularization**: every class receives the same effective regularization strength. In contrast, for Leaky-KL, the gradient of mini-batch is:

$$\frac{1}{KN} \sum_{n=1}^{KN} \frac{\partial \mathcal{L}_{\text{Leaky-KL}}^{(n)}}{\partial \alpha_i^{(n)}} = \frac{K-1}{K}(1 - r_i) \cdot \bar{g}_{Leak,i}, \quad (16)$$

where $r_i = n_i/(K-1)N$ is the *leak ratio* for class $i$, with $n_i$ denoting the number of non-target samples where class $i$ appears in the top-$k$ predictions. The effective regularization coefficient now becomes class-dependent:

$$\lambda_{\text{eff}}^{(i)} = \frac{K-1}{K}(1 - r_i). \qquad (17)$$

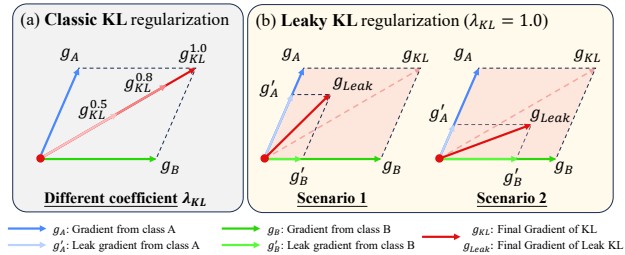

Figure 3. **Leaky-KL achieves a more flexible regularization adapted to data patterns.** Blue/green arrows represent gradients $\bar{g}_A$, $\bar{g}_B$ from classes A and B; red arrows show final regularization gradients. (a) Classic KL with varying $\lambda_{\text{KL}}$: gradients scale along a fixed direction, achieving only one-dimensional isotropic scaling. (b) Leaky-KL: different leak ratios produce different gradient directions, enabling two-dimensional anisotropic adaptation that spans the parallelogram region.

**Geometric visualization: From Scaling to Adaption.**
For simplicity, here we only consider two non-target classes A and B with gradients $\bar{g}_A$ and $\bar{g}_B$. In classic KL (Figure 3(a)), varying $\lambda_{\text{KL}}$ merely scales the combined gradient along a fixed direction composed by $\bar{g}_A$ and $\bar{g}_B$. This simple magnitude scaling also demonstrates the plateau effect:

when optimizing, there is no difference in constraint structure with varying coefficient. Also, it is just the reason why it fundamentally cannot escape the all-or-nothing dilemma. In contrast, Leaky-KL (Figure 3(b)) allows different classes to have different effective coefficients through varying leak ratios. When class A frequently appears in top-$k$ (higher $r_A$, Scenario 1), the gradient shifts toward the B-axis; when class B dominates (higher $r_B$, Scenario 2), it shifts toward the A-axis. By varying $(r_A, r_B)$ across different batches, the resultant gradients can span the entire parallelogram region (shaded) and access a two-dimensional solution space, leading to an adaptive constraint for the batch pattern.

In summary, coefficient tuning applies uniform scaling $\lambda_{\text{KL}}$ across all non-target classes, constituting isotropic regularization that merely adjusts constraint magnitude along a fixed direction. The batch-averaged gradient remains proportional to the same direction regardless of $\lambda_{\text{KL}}$ values, explaining the observed plateau effect. In contrast, Leaky-KL introduces class-dependent effective regularization strengths through top-$k$ masking, where frequently predicted classes receive reduced constraint (higher leak ratio) while rarely predicted classes receive stronger constraint (lower leak ratio). This anisotropic adaptation allows the regularization to dynamically respond to batch-specific evidence patterns, fundamentally transcending the one-dimensional magnitude scaling inherent in coefficient tuning.

## 4. Experiments

### 4.1. Experimental Setup

**Dataset Settings** To evaluate the effectiveness of Leaky-KL regularization and analyze the impact of the leak parameter $k$, we conducted experiments across three medical imaging modalities with distinct characteristics: endoscopy, dermatology, and histopathology. Following established protocols (Roy et al., 2022; Fu et al., 2025), we defined classes with relatively fewer samples as OOD to simulate rare disease scenarios in clinical practice. Additionally, for certain datasets, we incorporated image artifacts that may occur during acquisition to reflect the complexity of real-world clinical environments. Detailed dataset description and OOD class definitions are provided in Appendix B.1.

**Implementation Details** To ensure fairness, we used ResNet18 pre-trained on ImageNet as the backbone. All methods were fine-tuned for 15 epochs for endoscopy, 50 epochs for dermatology, and 100 epochs for histopathology. We adopted AdamW as the optimizer with a learning rate of 1e-4 and set $\lambda_{\text{KL}}$=1.0 as default for all methods. The experiments were performed with a batch size of 256 for dermatology and endoscopy, 512 for histopathology. All experiments are conducted using 5-fold cross-validation, and we report the mean and standard deviation across folds.

*Table 1.* **Comparison of OOD detection performance with state-of-the-art methods. Bold** indicates best performance per metric per dataset; underline indicates second-best. EDL with Leaky-KL achieves consistent superiority across all datasets.

| | Endoscopy | | | | Dermatology | | | | Histopathology | | | |
|---|---|---|---|---|---|---|---|---|---|---|---|---|
| | FPR@95 | AUROC | AUPR@IN | AUPR@OUT | FPR@95 | AUROC | AUPR@IN | AUPR@OUT | FPR@95 | AUROC | AUPR@IN | AUPR@OUT |
| EDL ($\lambda_{KL}$ = 1) | 59.68±3.04 | 90.12±0.40 | 59.76±1.60 | 97.36±0.14 | 87.16±7.45 | 70.30±2.73 | 81.48±2.27 | 49.93±2.93 | 79.67±1.05 | 81.77±0.26 | 72.68±0.38 | 85.27±0.25 |
| EDL ($\lambda_{KL}$ = 0) | 50.82±1.81 | 91.12±0.25 | 69.21±0.59 | 97.64±0.13 | 92.27±2.94 | 68.23±3.83 | 78.29±3.19 | 50.63±4.12 | 94.60±0.15 | 78.84±0.34 | 63.61±0.39 | 85.01±0.34 |
| EDL (optimal $\lambda_{KL}$) | 50.82±1.81 | 91.12±0.25 | 69.21±0.59 | 97.64±0.13 | 87.08±4.70 | 70.37±2.52 | 81.14±2.14 | 50.94±3.08 | 79.38±1.22 | 81.94±0.28 | 72.77±0.60 | 85.60±0.22 |
| RED [ICML'23] | 64.65±4.09 | 89.60±0.37 | 56.09±2.20 | 97.28±0.05 | 85.58±4.72 | 70.94±2.03 | 81.74±2.04 | 51.15±2.02 | 86.46±0.91 | 80.61±0.29 | 69.38±0.59 | 84.80±0.31 |
| I-EDL [ICML'23] | 73.23±3.85 | 88.33±0.88 | 55.06±2.08 | 96.99±0.27 | 86.18±3.33 | 69.62±2.58 | 80.99±2.04 | 50.02±3.79 | **74.35±1.09** | 82.18±0.14 | 74.66±0.34 | 85.31±0.20 |
| R-EDL [ICLR'24] | 61.37±3.22 | 89.32±0.94 | 66.11±2.50 | 97.08±0.33 | 85.79±4.81 | 70.28±2.68 | 81.48±2.23 | 49.93±3.13 | 77.30±1.30 | 82.19±0.28 | 73.45±0.59 | 85.78±0.27 |
| Re-EDL [TPAMI'25] | 61.58±4.09 | 89.72±0.54 | 63.09±3.06 | 97.28±0.18 | 87.65±1.04 | 69.54±0.98 | 80.10±0.46 | 52.02±1.76 | 82.35±2.85 | 79.90±0.86 | 68.90±1.41 | 84.48±0.52 |
| ERNN [MICCAI'23] | 51.82±2.46 | 90.91±0.25 | 68.83±1.75 | 97.57±0.12 | 92.14±2.99 | 68.25±2.99 | 78.43±2.29 | 50.61±3.94 | 94.80±0.23 | 78.62±0.39 | 63.28±0.36 | 84.85±0.43 |
| D-EDL [MedIA'25] | 48.71±2.06 | 91.24±0.21 | 71.46±1.09 | 97.58±0.11 | **84.97±2.52** | **72.19±1.10** | 81.01±0.92 | 54.50±1.33 | 78.49±0.96 | 82.16±0.27 | 72.59±0.49 | 86.12±0.23 |
| Leaky EDL (ours) | **35.95±2.37** | **92.32±0.55** | **77.97±0.86** | **97.71±0.28** | 88.40±2.35 | 71.94±2.10 | **81.80±1.67** | **54.75±2.17** | 74.63±3.17 | **83.94±0.38** | **76.83±0.58** | **87.59±0.35** |

**Evaluation Metrics**  Following OpenOOD (Yang et al., 2022), we evaluate OOD detection using AUROC, FPR@95, AUPR@IN, and AUPR@OUT. AUROC measures overall separability between ID and OOD distributions. FPR@95 reports the OOD false positive rate when the true positive rate for ID samples is 95%. AUPR@IN and AUPR@OUT measure the area under the precision-recall curve when ID and OOD are treated as the positive class, respectively. For ID classification performance, we report accuracy. Higher values indicate better performance for all metrics except FPR@95, where lower is better.

## 4.2. Comparison with State-of-the-Art Methods

We compare our method against classic EDL (Sensoy et al., 2018) and recent variants that address regularization design: RED (Pandey & Yu, 2023) and I-EDL (Deng et al., 2023) enhance regularization with Fisher Information, R-EDL (Chen et al., 2024) and Re-EDL (Chen et al., 2025) advocates relaxing non-essential settings in EDL including prior and KL regularization, while ERNN (Fu et al., 2023) and D-EDL (Fu et al., 2025) propose medical-specific adaptations. Moreover, we include an "optimal $\lambda_{KL}$" baseline representing the best performance achievable through coefficient tuning on each dataset (obtained via grid search over $\lambda_{KL} \in \{0, 0.25, 0.5, 0.75, 1.0\}$. For our method, we report results with the leak parameter fixed at $k=2$ across all datasets, demonstrating robust performance without dataset-specific tuning. As all methods achieve comparable ID classification accuracy (detailed in Appendix B.2), Table 1 focuses exclusively on OOD detection performance.

**Distinct KL Response Patterns Across Datasets.**  As shown in Table 1, the two KL extremes exhibit distinct dataset-dependent response patterns, empirically validating the all-or-nothing dilemma (Section 2.2). On Endoscopy, removing constraint ($\lambda_{KL}=0$) substantially outperforms full constraint ($\lambda_{KL}=1$), improving AUROC from 90.12% to 91.12% and AUPR@IN from 59.76% to 69.21%. In contrast, both Dermatology and Histopathology favor full constraint: on Dermatology, AUROC reaches 70.30% versus

68.23%, while on Histopathology, AUROC achieves 81.77% versus 78.84% with AUPR@IN showing substantial degradation from 72.68% to 63.61% when constraint is removed. This dataset-dependent behavior confirms that neither extreme provides universal reliability across diverse medical imaging scenarios.

**Consistent Superior Performance Across Datasets.**  In stark contrast to the dataset-dependent behavior of KL extremes, Leaky-KL demonstrates robust superior performance across all three datasets with a single fixed hyperparameter ($k=2$). On Endoscopy, our method achieves the best results across all metrics (AUROC: 92.32%, FPR@95: 35.95%). On Histopathology, we obtain top performance with AUROC of 83.94% and AUPR@IN of 76.83%. Even on Dermatology, our method maintains competitive performance (AUROC: 71.94%, ranking second) while achieving the best AUPR@IN of 81.80%. Crucially, Leaky-KL consistently surpasses the dataset-specific optimal coefficient tuning baseline across all datasets, demonstrating that anisotropic adaptation accesses performance regions fundamentally unreachable through isotropic coefficient scaling without requiring dataset-specific hyperparameter tuning.

## 4.3. Ablation on Regularization Strength Adjustment

To systematically investigate how regularization strength affects OOD detection, we conduct comprehensive ablation studies comparing coefficient tuning (varying $\lambda_{KL}$) and leakage adjustment (varying $k$). Figure 4 presents detailed comparisons across three datasets and four metrics.

**Breaking the Plateau Through Anisotropic Adaptation.** Coefficient tuning exhibits pronounced plateau effects: performance remains nearly constant across all non-zero $\lambda_{KL}$ values, with abrupt transitions only at $\lambda_{KL}=0$. This binary behavior is evident across all datasets, with curves showing minimal variation across $\lambda_{KL} \in [0.25, 1.0]$ before changing sharply at zero. In stark contrast, leakage adjustment demonstrates smooth, continuous variations. On Endoscopy, metrics peak at $k=2$ before stabilizing at $k=4$-6.

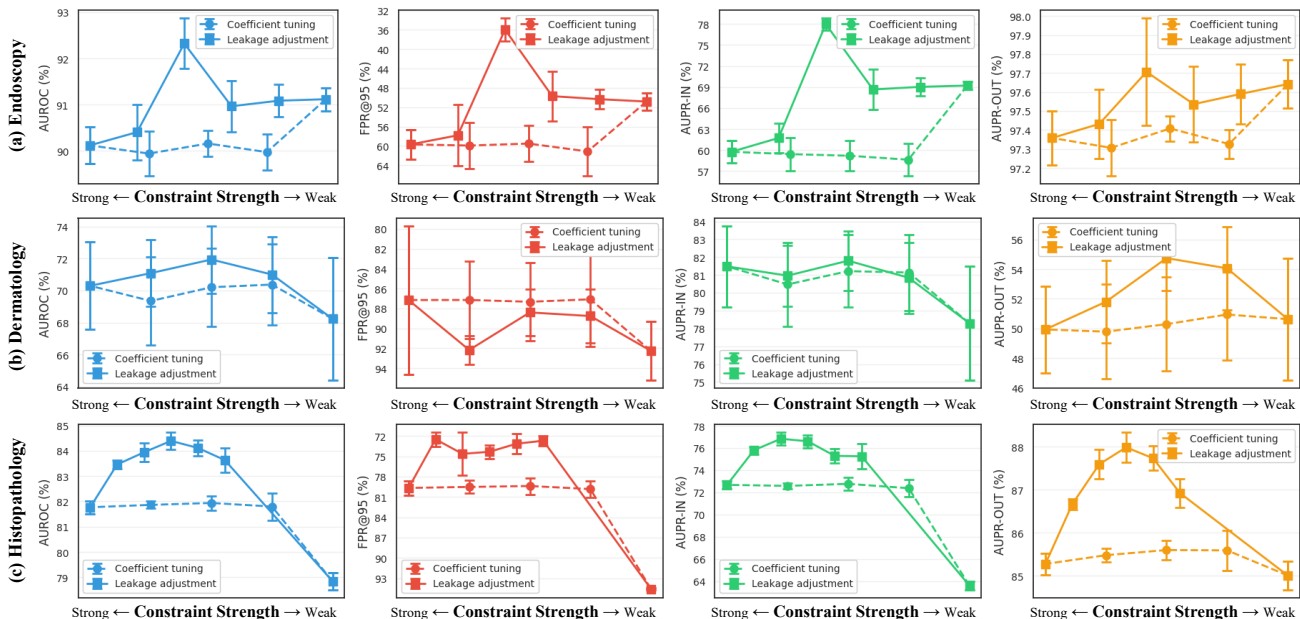

*Figure 4.* **Comparison of Leaky-KL and coefficient tuning for varying constraint strengths.** Coefficient tuning (dashed lines, varying $\lambda_{\mathrm{KL}}$) exhibits plateau effects with performance stable across non-zero values and abrupt transitions at 0. Leaky-KL (solid lines, varying $k$) demonstrates smooth performance variations and consistently matches or exceeds coefficient tuning across the constraint spectrum.

On Histopathology, clear monotonic trends emerge. Even on Dermatology's AUPR@IN, which shows a brief plateau at $k$=1-3, other metrics vary smoothly. This validates our gradient analysis (Section 3.2): coefficient tuning confines optimization to one-dimensional isotropic scaling, while leakage adjustment enables two-dimensional anisotropic adaptation beyond the all-or-nothing extremes.

**Systematic Superiority Across Constraint Spectrum.** Leakage adjustment systematically outperforms coefficient tuning across the regularization spectrum. On Endoscopy, our method consistently exceeds coefficient tuning across all four metrics. On Histopathology, leakage adjustment substantially surpasses coefficient tuning across all evaluated $k$ values (leakage ablation evaluation stops at $k$=5 as trends become evident). On Dermatology, while FPR@95 favors coefficient tuning at the 95% TPR operating point, the three distribution-characterizing metrics (AUROC, AUPR@IN, AUPR@OUT) all demonstrate leakage superiority. Overall, leakage achieves comprehensive superiority on two datasets while maintaining strong performance on three of four metrics on Dermatology. Crucially, peak performance systematically surpasses optimal coefficient tuning on all datasets (Table 1), demonstrating that anisotropic adaptation accesses regions fundamentally unreachable through isotropic scaling.

**Robust Default Setting.** Despite substantial dataset variations, we empirically find that $k$=2 consistently emerges as the optimal or near-optimal choice across all three datasets.

### 4.4. Dual Improvements in Both ID and OOD Detection

Effective OOD detection requires both identifying OOD samples with high uncertainty and maintaining confident predictions on ID samples. We analyze how different regularization strategies navigate this trade-off through quantitative metrics and distribution visualizations.

**Quantitative Analysis of Dual Improvements.** As shown in Table 1, it reveals a critical distinction that most existing EDL variants predominantly enhance AUPR@IN while leaving AUPR@OUT largely unchanged, whereas Leaky-KL achieves balanced improvements with dataset-specific patterns. On Endoscopy, where AUPR@OUT already reaches high values, Leaky-KL focuses improvement on AUPR@IN. On Dermatology and Histopathology, both metrics improve substantially: on Dermatology, AUPR@OUT increases notably where most methods show minimal gains, while on Histopathology, simultaneous improvements are observed in both metrics. Moreover, Figure 4 demonstrates that this dual improvement persists across the entire constraint spectrum, not merely at the optimal $k$ value. Throughout the regularization range, leakage adjustment consistently outperforms or matches coefficient tuning in both metrics across all three datasets. This systematic advantage validates our mechanism: selective class-wise leakage preserves low uncertainty for ID samples while constraining spurious evidence on OOD samples, whereas isotropic methods cannot differentiate between plausible alternatives and spurious evidence.

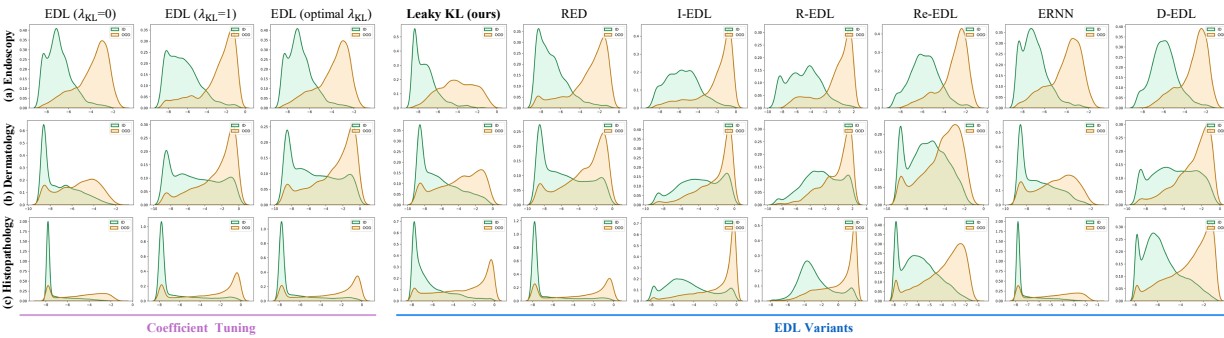

*Figure 5.* **Uncertainty Distributions of coefficient tuning and EDL variants.** Green and orange curves represent ID and OOD uncertainty distributions, respectively. Greater separation between ID and OOD distributions indicates better detection performance. Leaky-KL achieves clear separation while maintaining tight ID distributions, demonstrating effective balance between preserving ID confidence and constraining OOD uncertainty.

**Visualization of Density Distributions.** To get a more instinctive understanding of the dual improvement, Figure 5 visualizes uncertainty score distributions for ID (green) and OOD (orange) samples, where greater separation indicates better detection performance (full results in Appendix B.3). Compared to the unconstrained baseline (EDL with $\lambda_{KL}=0$), all regularized methods successfully shift OOD distributions toward higher uncertainty regions. However, methods differ substantially in their effect on ID uncertainty. Strong uniform constraint (EDL with $\lambda_{KL}=1$) produces wider, less confident ID distributions with flattened peaks, stemming from indiscriminately penalizing all non-target classes including plausible alternatives on ambiguous ID samples. In contrast, Leaky-KL demonstrates sharp, concentrated ID peaks alongside well-separated OOD distributions across all three datasets. On Endoscopy and Histopathology, our method achieves the tightest ID distributions with clear OOD separation, while even on Dermatology where subtle visual similarities make the task more challenging, Leaky-KL maintains concentrated ID distributions with improved OOD separation. This visualization confirms that anisotropic regularization achieves the optimal trade-off between ID confidence preservation and OOD uncertainty constraint, providing visual validation of the quantitative improvements.

### 4.5. Limitations and Future Directions

Although Leaky-KL successfully mitigates the dataset-dependent behavior of existing EDL methods through sample-adaptive regularization, several aspects of our framework require further investigation.

**More flexible regularization.** Our approach applies a fixed leak ratio $k$ uniformly across all samples within a dataset, however, it does not achieve true sample-adaptive regularization. Different samples exhibit varying degrees of ambiguity: clear-cut cases may require minimal constraint, while ambiguous samples near decision boundaries could

benefit from stronger regularization. Dynamically adjusting $k$ based on sample-specific characteristics, such as prediction confidence or proximity to decision boundaries, could enable the regularization to naturally adapt to each sample's inherent ambiguity, moving from dataset-level to truly sample-level anisotropic adaptation.

**Metric sensitivity to ID-OOD imbalance.** Our datasets exhibit varying ID-OOD imbalance ratios. We observe that AUROC, as a threshold-independent metric, is disproportionately influenced by the performance on the majority class. When ID samples dominate, AUROC tracks more closely with AUPR@IN, while AUPR@OUT improvements contribute less to overall AUROC. This imbalance-dependent metric behavior suggests that evaluation on datasets with different ID-OOD distributions may emphasize different performance aspects. Future work could benefit from either rebalanced evaluation protocols or alternative metrics that are less sensitive to class distribution, providing more robust assessment of OOD detection methods.

## 5. Conclusion

In this paper, we identified and addressed the all-or-nothing dilemma in Evidential Deep Learning, where existing methods apply uniform constraint either to all non-target classes or to none, leading to dataset-dependent performance. We revealed that this stems from the isotropic nature of coefficient tuning, which merely performs magnitude scaling without escaping the binary all-or-nothing framework. To address this, our proposed Leaky-KL regularization achieves anisotropic adaptation through selective class-wise leakage, enabling directional adjustment in gradient space that fundamentally transcends one-dimensional magnitude scaling. Comprehensive experiments demonstrate that Leaky-KL with $k=2$ consistently outperforms existing EDL variants and surpasses optimal dataset-specific coefficient tuning, proving it a principled solution for robust OOD detection.

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

# A. Related works

## A.1. Evidential Deep Learning

Among various uncertainty quantification approaches for deep learning, Evidential Deep Learning (EDL) (Sensoy et al., 2018) has gained increasing attention in safety-critical applications (Bao et al., 2021; Chen et al., 2023; Capellier et al., 2019; Zhou et al., 2023; Sun et al., 2025) due to its ability to provide single-pass uncertainty quantification with training stability and computational efficiency. Grounded in subjective logic theory (Jøsang, 2016), EDL models classification outcomes as Dirichlet distributions over class probabilities, enabling explicit quantification of epistemic uncertainty without requiring ensemble methods or multiple inference passes.

Subsequent work has extended EDL along various aspects, including improving OOD robustness with explicit regularization (Zhao et al., 2019), incorporating Fisher Information Matrix for better uncertainty estimation (Deng et al., 2023), and addressing imbalanced data through distributionally robust optimization (Sapkota & Yu, 2023). Despite these diverse improvements, all variants retain the core KL regularization component—a design choice that remained unquestioned until recently challenged by Re-EDL.

## A.2. The KL Regularization Debate

A central component of EDL is the KL divergence regularization term, originally introduced to constrain evidence accumulation on non-target classes. The original EDL formulation applies this regularization uniformly across all incorrect classes, with the motivation that restricting evidence for non-target categories would encourage the model to express higher uncertainty when it cannot confidently classify a sample. This design has been widely adopted across EDL variants as a fundamental mechanism for uncertainty quantification.

Recent work has begun to challenge this design. Re-EDL (Chen et al., 2025) argues that KL regularization is "nonessential" and proposes its complete removal to improve OOD detection. Through systematic experiments, they demonstrate that removing KL regularization while relaxing prior weight constraints achieves superior performance on several benchmark datasets. While they acknowledge the potential benefit of coefficient tuning, they ultimately recommend zero regularization for its simplicity and broad applicability. Their analysis suggests that KL regularization causes information loss in evidence magnitude, thereby hindering the model's ability to distinguish between in-distribution and out-of-distribution samples. Beyond Re-EDL, several other recent works including ERNN (Fu et al., 2023), D-EDL (Fu et al., 2025), and H-EDL (Qu et al., 2024) have also reduced or eliminated KL regularization to achieve improved performance, lending further support to this perspective.

However, this "no constraint" approach overlooks scenarios where constraint remains essential. Our analysis reveals that both extremes—full constraint and no constraint—exhibit dataset-dependent behavior, with each succeeding on some datasets while failing on others. Recent work D-EDL takes a different approach by demonstrating that appropriate constraint remains beneficial, but achieves this through uniform evidence transformation on the classification loss. This indirect approach may inadvertently affect classification performance and lacks the flexibility to provide selective constraint. The question of how to directly design anisotropic KL regularization that adapts to different samples remains open.

## A.3. Medical Out-of-Distribution Detection

Out-of-distribution detection in medical imaging presents unique challenges due to high class imbalance, subtle inter-class variations, and diverse OOD types ranging from unseen pathological conditions to image quality issues. Despite the critical importance of reliable uncertainty quantification in safety-critical medical applications, research applying EDL-based methods to medical OOD detection remains limited. While recent works such as ERNN (Fu et al., 2023), D-EDL (Fu et al., 2025) have demonstrated the potential of evidential learning on medical datasets, systematic investigation of how regularization strategies perform across diverse medical imaging modalities is lacking. This gap is significant because, as we demonstrate, medical datasets can exhibit markedly different characteristics that lead to dataset-dependent performance of existing methods. We address this challenge through a unified anisotropic regularization framework that achieves robust performance across diverse medical imaging scenarios.

# B. Experiment Supplementaries.

## B.1. Dataset Settings

We evaluate our method across three medical imaging datasets spanning different modalities, scales, and clinical applications. These datasets exhibit substantial variation in scale (8K, 25K, 171K images), modalities, and distributional shifts. Figure 6 shows representative samples and Table 2 provides detailed statistics.

*Table 2.* Dataset distributions for in-distribution (ID) and out-of-distribution (OOD) classes across three medical imaging datasets.

| (a) Endoscopy Dataset | | | | | | | | ALL |
|---|---|---|---|---|---|---|---|---|
| **ID** | z-line | retroflex-s | pylorus | cecum | polyps | | | 4732 |
| | 932 | 764 | 999 | 1009 | 1028 | | | |
| **OOD** | esophagitis | retroflex-r | ulcerative-colitis | hemorrhoids | ileum | barretts | bbps-0-1 | EAD-artifact | 3986 |
| | 663 | 391 | 851 | 6 | 9 | 94 | 646 | 1326 | |

| (b) Dermatology Dataset | | | | ALL |
|---|---|---|---|---|
| **ID** | MEL | NV | BCC | BKL | 23344 |
| | 4522 | 12875 | 3323 | 2624 | |
| **OOD** | DF | VASC | AK | SCC | 1987 |
| | 239 | 253 | 867 | 628 | |

| (c) Histopathology Dataset | | | | | | | | | ALL |
|---|---|---|---|---|---|---|---|---|---|
| **ID** | BLA | EBO | EOS | LYT | MYB | NGB | NGS | PLM | PMO | 137064 |
| | 11973 | 27395 | 5883 | 26242 | 6557 | 9968 | 29424 | 7629 | 11993 | |
| **OOD** | ART | NIF | KSC | ABE | BAS | FGC | HAC | LYI | MMZ | 34390 |
| | 19630 | 3538 | 42 | 8 | 441 | 47 | 409 | 65 | 3055 | |
| | MON | OTH | PEB | | | | | | | |
| | 4040 | 294 | 2740 | | | | | | | |

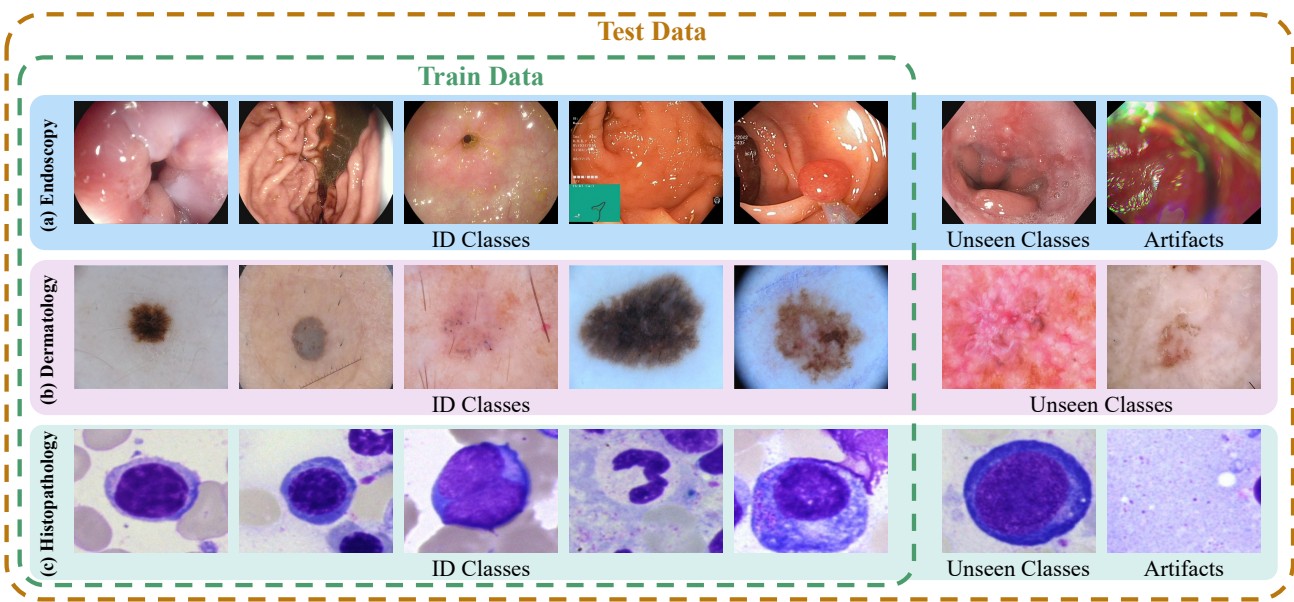

*Figure 6.* Representative samples of ID and OOD from the three medical imaging datasets.

**Endoscopy**  We use the Hyper-Kvasir (HK) dataset (Borgli et al., 2020) and the Endoscopy Artifact Detection (EAD) dataset (Ali et al., 2019; 2021; 2020b;a), totaling 8,718 images. Classes with fewer than 500 samples from HK are designated as rare-class OOD. Additionally, we incorporate artifact images from EAD as artifact-based OOD, where images with more than 25 artifact targets or more than 40% artifact area are selected. This dual strategy evaluates both semantic shifts (rare anatomical structures) and perceptual shifts (imaging quality degradation).

**Dermatology**  We use the ISIC2019 dataset (Tschandl et al., 2018; Codella et al., 2018; Combalia et al., 2019) comprising dermoscopic images across 8 skin lesion categories. Classes with fewer than 1,000 samples are designated as rare-class OOD, while remaining classes serve as ID data. This dataset focuses solely on rare-class OOD detection without artifact samples.

**Histopathology**  We use the Bone Marrow Cytomorphology (BMC) dataset (Matek et al., 2021b;a) containing expert-annotated images of bone marrow cells from a leukemia diagnostic laboratory. Classes with fewer than 5,000 samples are designated as rare-class OOD. The dataset also includes an artifact class capturing technical imaging issues, providing both rare cell types and artifact-based OOD samples.

### B.2. In-Distribution Classification Accuracy

Table 3 presents the in-distribution classification accuracy across all methods and datasets. All methods achieve comparable performance, with accuracy ranging from 99.51%–99.84% on Endoscopy, 87.17%–87.73% on Dermatology, and 91.36%–91.56% on Histopathology. The negligible differences in ID accuracy (within 0.4% across methods) confirm that improvements in OOD detection are not due to better feature representations or classification capability, but rather from more effective uncertainty quantification. This validates our focus on OOD detection performance in the main results.

*Table 3.* In-distribution classification accuracy (%) across all methods and datasets..

| Accuracy | Endoscopy | Dermatology | Histopathology |
|---|---|---|---|
| EDL ($\lambda_{KL} = 1$) | 99.54±0.27 | 87.17±0.60 | 91.40±0.09 |
| EDL ($\lambda_{KL} = 0$) | 99.58±0.23 | 87.42±0.94 | 91.26±0.11 |
| RED [ICML'23] | 99.58±0.25 | 87.71±0.52 | 91.36±0.06 |
| I-EDL [ICML'23] | 99.58±0.15 | **87.90±0.60** | 91.45±0.05 |
| R-EDL [ICLR'24] | **99.60±0.14** | 87.22±0.85 | 91.36±0.06 |
| Re-EDL [TPAMI'25] | 99.54±0.24 | 87.55±0.45 | 91.17±0.13 |
| ERNN [MICCAI'23] | 99.54±0.24 | 87.27±0.42 | 91.36±0.07 |
| D-EDL [MedIA'25] | 99.54±0.24 | 87.47±0.74 | 91.13±0.06 |
| Leaky EDL (ours) | 99.51±0.16 | 87.60±0.58 | **91.56±0.05** |

### B.3. Visualization of Density Distributions

Figure 7, 8, and 9 visualize the uncertainty score distributions for ID and OOD samples across all methods and datasets under 5-fold cross-validation. Green curves represent ID samples, while orange curves represent OOD samples. Better OOD detection corresponds to greater separation between the two distributions.

Our Leaky KL method consistently achieves clear separation between ID and OOD distributions across all three datasets, comparable to or better than baseline methods. Notably, on Endoscopy and Histopathology where standard EDL variants show significant overlap (especially EDL with $\lambda_{KL} = 0$), our method maintains robust separation. This visualization confirms the quantitative results in Table 1, demonstrating that anisotropic regularization effectively balances ID confidence and OOD uncertainty without the distribution collapse issues observed in some baselines.

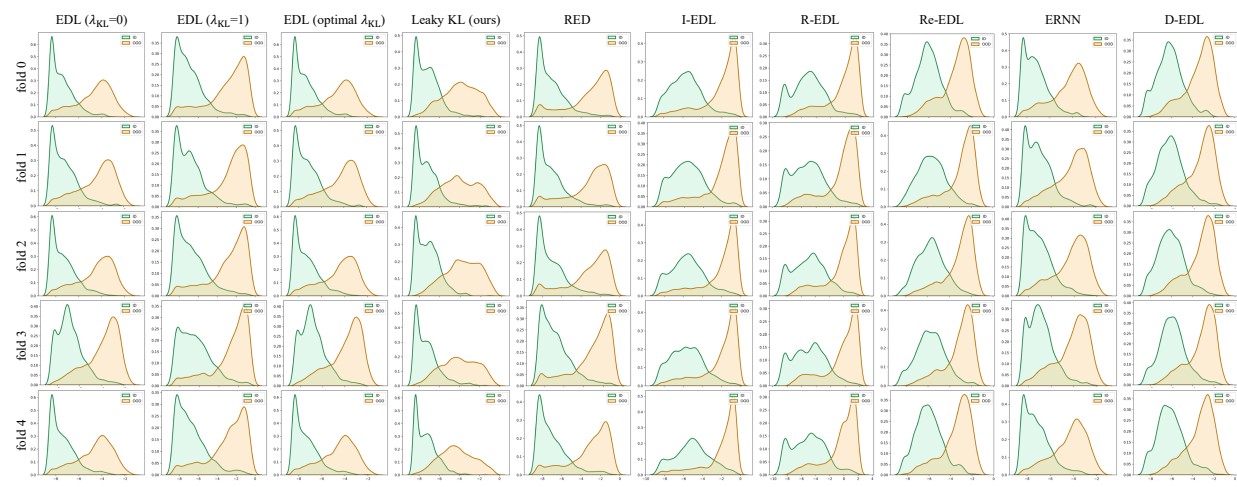

*Figure 7.* **Uncertainty score distributions on Endoscopy dataset across 5-fold cross-validation.** Green curves represent ID samples and orange curves represent OOD samples. Better separation indicates more effective OOD detection.

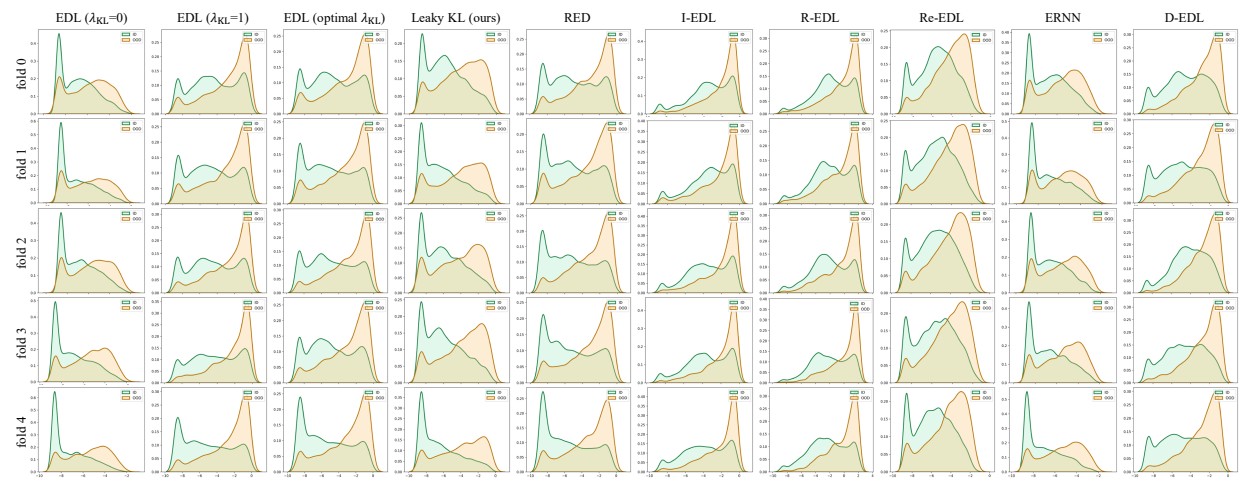

*Figure 8.* **Uncertainty score distributions on Dermatology dataset across 5-fold cross-validation.** Green curves represent ID samples and orange curves represent OOD samples.

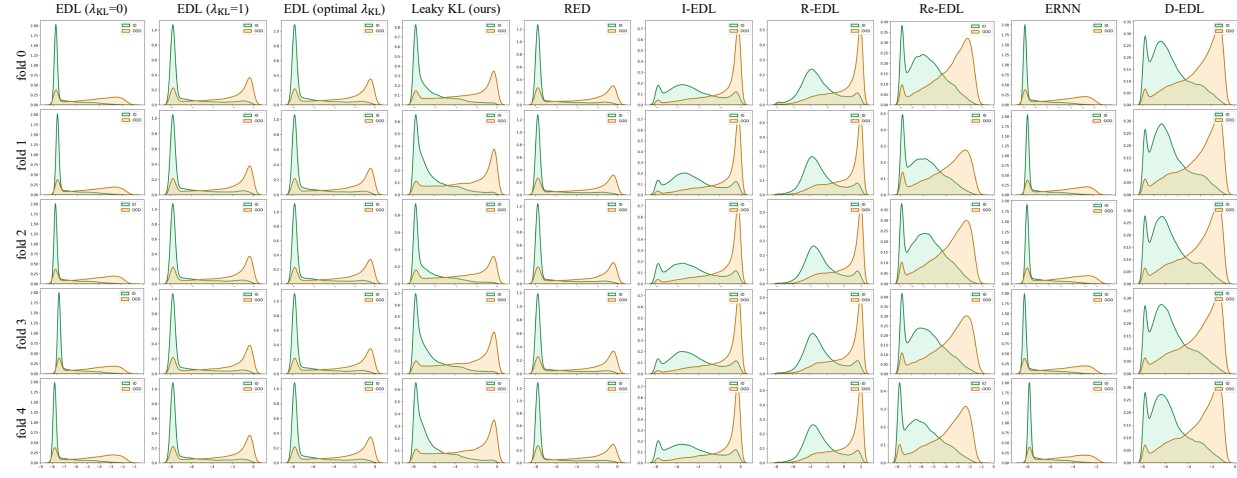

*Figure 9.* **Uncertainty score distributions on Histopathology dataset across 5-fold cross-validation.** Green curves represent ID samples and orange curves represent OOD samples.

