# OpenReview forum: "Beyond All-or-Nothing: Anisotropic Evidential Regularization for Robust Medical Out-of-Distribution Detection"
_ICML.cc/2026/Conference — Submitted to ICML 2026_

### Official Review · Reviewer_KPyK · 2026-03-10

**Soundness:** 3
**Presentation:** 3
**Significance:** 2
**Originality:** 2
**Overall Recommendation:** 3
**Confidence:** 3

**Summary:**

This paper studies out-of-distribution detection in medical imaging under the Evidential Deep Learning (EDL) framework. The authors argue that existing KL regularization faces an “all-or-nothing” dilemma: fully constraining all non-target classes can hurt ambiguous in-distribution samples, while removing the constraint can make OOD samples overconfident. To address this, they propose Leaky-KL, which exempts the top-k evidence classes from KL regularization and keeps the remaining classes constrained. Experiments on three medical datasets show that this simple selective regularization strategy consistently improves OOD detection over prior EDL variants and standard KL coefficient tuning.

**Compliance With Llm Reviewing Policy:**

Affirmed.

**Final Justification:**

Since the key hyperparameter k appears to be selected heuristically rather than learned or systematically justified, I am inclined to maintain my original score.

**Key Questions For Authors:**

1. Please clarify how k and the “optimal λ” baseline were selected. Were they chosen independently on training/validation data within each fold, or did the selection use test-fold information? Is there any test-time tuning？

2. How robust is Leaky-KL across different backbones, OOD types, and non-medical datasets? This is especially important since the current evaluation uses only a single backbone and three medical datasets. Additional results on diverse architectures would significantly strengthen my assessment of the paper’s significance and generalizability.

3. How should k=2 be chosen relative to the number of classes or dataset properties? The paper shows that k=2 is empirically robust, but the principle remains unclear. A more systematic rule or an adaptive-k discussion would improve the practical value of the method.

4. Can the anisotropic regularization claim be supported more directly? For example, showing empirical leak ratios, class-wise gradient statistics, or additional analysis linking these quantities to performance would strengthen the theoretical story.

**Limitations:**

yes

**Strengths And Weaknesses:**

**Strengths.**

(1) Soundness: The paper is well motivated by an empirical observation: the two KL extremes behave differently across datasets. The experimental setup is reasonably careful, with a unified ResNet18 backbone, multiple OOD metrics, 5-fold cross-validation, and comparisons against classic EDL, Re-EDL, RED, I-EDL, R-EDL, ERNN, D-EDL, and an optimal-λ tuning baseline.

(2) Presentation: The narrative is easy to follow. The paper starts from the all-or-nothing dilemma, shows opposing behaviors of KL extremes, and then motivates Leaky-KL. Figures 1–5 and Table 1 support the central story effectively.

(3) Significance: Medical OOD detection is an important problem in safety-critical settings. The method shows strong performance across three medical modalities, suggesting practical value for EDL-based medical OOD detection.

(4) Originality: The method is simple, but the shift from “how much to constrain” to “what to constrain” is a meaningful contribution. Selective leakage is more novel than merely retuning λ.


**Weaknesses.**

(1) Soundness: The theoretical section is more of an intuitive gradient-based explanation than a formal guarantee; the anisotropic-regularization claim is supported mainly by analysis intuition and experiments. A key concern is that the paper should more clearly explain how k and the “optimal λ” baseline were selected within cross-validation; if test information was used, the fairness of the comparison would be weakened.

(2) Presentation: A clearer algorithm box or pseudocode would also improve reproducibility.

(3) Significance: The evaluation scope is still limited to three medical datasets and one backbone, so it remains unclear how well the conclusion generalizes to broader OOD settings, non-medical benchmarks, or stronger backbone architectures.

(4) Originality: The idea is novel, but technically it is still a lightweight modification of the KL regularizer within EDL, rather than a fundamentally new uncertainty framework. Its novelty is best viewed as an insightful and effective design improvement.

---

> ### Author Rebuttal · Authors · 2026-03-29
>
> Dear Reviewer KPyK,
>
> We sincerely appreciate the time and effort you have dedicated to reviewing our paper and providing us with valuable feedback. Here are our replies.
>
> > Q1: Please clarify how k and the "optimal λ" baseline were selected.
>
> We clarify that in Leaky-KL k=2 is a fixed empirical parameter selected without any test-fold information. The optimal λ baseline is deliberately designed as an oracle upper bound — selecting the best-performing λ∈{0, 0.25, 0.5, 0.75, 1.0} on the test set to give isotropic tuning the most favorable possible conditions. The fact that Leaky-KL with a fixed k=2 still consistently outperforms this oracle directly confirms our central claim: anisotropic regularization accesses a solution space that isotropic coefficient tuning cannot reach, regardless of how well the coefficient is chosen.
>
> > Q2: How should k=2 be chosen relative to the number of classes or dataset properties?
>
> We acknowledge that k=2 is an empirical choice, but offer the following rationale. When k=1, the exempted class only allows the model to "preserve its own prediction" without capturing inter-class correlations; as predictions converge to the correct class during training, the top-1 class exemption increasingly overlaps with the target mask, leading to a degradation from Leaky-KL to classic KL regularization. k=2 is therefore the most conservative leak that meaningfully models class correlations while minimizing relaxation of OOD constraints, and empirically demonstrates stable and favorable performance across all three medical datasets. Nevertheless, as shown in Figure 4, k=3 outperforms k=2 on Histopathology, suggesting that adaptive-k selection remains an important direction for future work.
>
> > Q3: How robust is Leaky-KL across different backbones, OOD types, and non-medical datasets?
>
> For dataset generalizability, we provide additional experiments on natural image benchmarks in our response to Reviewer se9V, Q2, and a comprehensive comparison with non-EDL baselines in our response to Reviewer 7nkc, Q3, where Leaky-KL achieves state-of-the-art performance among EDL variants across both medical and natural image domains. Regarding backbone robustness, EDL-based uncertainty estimation operates at the loss level, making the contribution orthogonal to architecture choice.
>
> > Q4: Can the anisotropic regularization claim be supported more directly?
>
> We thank the reviewer for this valuable suggestion. We provide two analyses using a Leaky-KL trained model on the Bone Marrow dataset (9 classes) to directly support the anisotropic regularization claim.
>
> First, the table below shows per-class leak ratios at k=2 computed on both training and validation ID data. The high agreement across splits (Spearman ρ=0.98, p<0.001) confirms that leak ratios reflect intrinsic data properties stable across different data splits.
>
> |Class|Train|Val_ID|Diff|
> |-|-|-|-|
> |BLA|0.0522|0.0627|+0.0105|
> |EBO|0.2626|0.2576|-0.0049|
> |EOS|0.0023|0.0028|+0.0005|
> |LYT|0.3237|0.3207|-0.0030|
> |MYB|0.0752|0.0788|+0.0036|
> |NGB|0.2318|0.2337|+0.0019|
> |NGS|0.1493|0.1509|+0.0017|
> |PLM|0.0049|0.0116|+0.0068|
> |PMO|0.0770|0.0780|+0.0010|
>
> Second, the cross-k leak ratio table below shows how leak ratios vary across classes and k values. As described in Section 3.2, different k values induce different effective scaling coefficients per class. EOS (morphologically distinctive, leak ratio 0.0028) and LYT (high intra-class variability, leak ratio 0.3207) exemplify the class-dependent variation that isotropic tuning cannot provide.
>
> |Class|k=0|k=1|k=2|k=3|k=4|k=5|k=6|k=7|k=8|k=9|
> |-|-|-|-|-|-|-|-|-|-|-|
> |BLA|0.0000|0.0121|0.0627|0.1968|0.2644|0.3276|0.3929|0.4829|0.6215|1.0000|
> |EBO|0.0000|0.0088|0.2576|0.3959|0.4907|0.5605|0.6228|0.6955|0.8173|1.0000|
> |EOS|0.0000|0.0008|0.0028|0.1123|0.2196|0.3438|0.4917|0.6791|0.8922|1.0000|
> |LYT|0.0000|0.0149|0.3207|0.4719|0.5822|0.6638|0.7394|0.8247|0.9197|1.0000|
> |MYB|0.0000|0.0069|0.0788|0.2333|0.4567|0.6552|0.7961|0.8877|0.9491|1.0000|
> |NGB|0.0000|0.0130|0.2337|0.2981|0.3899|0.5098|0.6597|0.8238|0.9481|1.0000|
> |NGS|0.0000|0.0220|0.1509|0.2893|0.4306|0.5613|0.6917|0.8062|0.9149|1.0000|
> |PLM|0.0000|0.0023|0.0116|0.1181|0.2335|0.3694|0.5334|0.7070|0.8717|1.0000|
> |PMO|0.0000|0.0173|0.0780|0.2023|0.3663|0.5515|0.7228|0.8519|0.9394|1.0000|
>
> > Q5: The theoretical section is more of an intuitive gradient-based explanation than a formal guarantee. A clearer algorithm box or pseudocode would also improve reproducibility.
>
> While a formal theoretical guarantee remains an open challenge in the EDL framework, the gradient-based analysis in Section 3.2, combined with the empirical leak ratio results above, provides a coherent explanation of why anisotropic regularization improves OOD detection. A pseudocode comparison of Standard KL and Leaky-KL will be included in the camera-ready version to improve reproducibility.

---

> > ### Author Rebuttal · Reviewer_KPyK · 2026-04-03
> >
> > Setting the key parameter k = 2 is overly empirical, so I will keep my score unchanged.

---

> > > ### Author Response · Authors · 2026-04-03
> > >
> > > Dear Reviewer KPyK,
> > >
> > > Thank you for acknowledging that your concerns have been addressed. We would like to offer one clarification regarding k=2.
> > >
> > > Our central contribution is the leakage adjustment mechanism itself, not the specific value of k. As shown in Figure 4, across the entire constraint spectrum, leakage adjustment consistently matches or outperforms coefficient tuning at every comparable constraint strength—this systematic superiority holds *regardless of which k is selected*. k=2 serves simply as a practical default that requires no dataset-specific tuning; the advantage of anisotropic regularization over isotropic scaling is not contingent on this particular choice.
> > >
> > > We appreciate the constructive engagement throughout the review process.

---

### Official Review · Reviewer_se9V · 2026-03-11

**Soundness:** 3
**Presentation:** 3
**Significance:** 2
**Originality:** 2
**Overall Recommendation:** 3
**Confidence:** 3

**Summary:**

The paper focuses on regularization in EDL for OOD detection in medical imaging and proposes Leaky-KL regularization, an anisotropic evidential regularization method.

**Compliance With Llm Reviewing Policy:**

Affirmed.

**Final Justification:**

I would prefer to keep my rating unchanged.

**Key Questions For Authors:**

See weaknesses.

**Limitations:**

yes

**Strengths And Weaknesses:**

Pros:
- The paper is well organized and generally easy to follow.
- The authors point out a concrete limitation in existing approaches, and the proposed Leaky-KL regularization is conceptually simple and easy to understand.

Cons:
- The proposed method mainly modifies the KL regularization with a top-k leakage mechanism. Overall, the conceptual advance seems somewhat modest to me.
- All experiments are conducted on medical datasets, so it is unclear whether the method would generalize well to broader OOD detection settings.
- The method introduces a new hyperparameter (top-k), but there is limited discussion about how sensitive the performance is to this choice. For example, I cannot find the k values in Figure 4.
- The evaluation mainly compares against other EDL-based methods. Including comparisons with non-EDL OOD detection approaches would make the empirical claims stronger.
- Although the intuition is fairly clear, the paper provides limited theoretical analysis explaining why anisotropic regularization should improve OOD detection.

---

> ### Author Rebuttal · Authors · 2026-03-29
>
> Dear Reviewer se9V,
>
> We sincerely appreciate the time and effort you have dedicated to reviewing our paper and providing us with valuable feedback. Here are our replies.
>
> > Q1: The proposed method mainly modifies the KL regularization with a top-k leakage mechanism. Overall, the conceptual advance seems somewhat modest to me.
>
> The core contribution lies not in the top-k mechanism itself, but in identifying a structural limitation of isotropic KL regularization and establishing the necessity of shifting from scaling to direction as the key degree of freedom. As shown in Figure 4, coefficient tuning behaves not as a continuous knob but effectively as a switch: once λ>0, performance plateaus regardless of the λ value, confirming that the limitation is structural. Within the KL framework, direction, not magnitude, is therefore the only remaining degree of freedom. Leaky-KL is a clean instantiation of this direction, whose simplicity isolates the contribution of anisotropic regularization itself.
> > Q2: It is unclear whether the method would generalize well to broader OOD detection settings.
>
> We provide additional experiments on standard OOD benchmarks using CIFAR and MNIST as in-distribution datasets, evaluated against multiple OOD datasets including SVHN, Textures, CIFAR, MNIST, and Place365 in the table below. Although R-EDL achieves lower FPR@95 on CIFAR10→MNIST, this comes at the cost of a substantial AUROC drop (75.19 vs. 86.28), reflecting an unfavorable trade-off. Leaky-KL achieves the best overall performance among EDL-based methods across both in-distribution settings, confirming its generalizability beyond the medical domain.
> ||CIFAR10|→MNIST|CIFAR10|→Texture|CIFAR10|→SVHN|\||MNIST|→CIFAR10|MNIST|→Texture|MNIST|→Place365|
> |-|-|-|-|-|-|-|-|-|-|-|-|-|-|
> |Method|FPR@95|AUROC|FPR@95|AUROC|FPR@95|AUROC|\||FPR@95|AUROC|FPR@95|AUROC|FPR@95|AUROC|
> |EDL(λ=0)|87.84|81.32|78.09|84.13|42.84|83.75|\||3.37|98.80|10.68|96.75|5.93|98.10|
> |EDL(λ=0.25)|51.46|80.62|50.60|82.21|43.44|81.60|\||8.19|97.86|55.30|89.35|9.24|97.43|
> |EDL(λ=0.5)|49.10|79.97|46.49|82.87|42.17|82.22|\||8.90|97.32|55.76|88.98|9.68|97.14|
> |EDL(λ=0.75)|38.68|85.85|48.72|82.27|39.83|81.19|\||8.52|97.44|55.33|89.53|8.57|97.30|
> |EDL(λ=1.0)|46.97|83.78|46.27|82.15|39.04|82.65|\||8.03|97.67|53.79|90.78|9.43|97.31|
> |RED|56.70|79.99|53.16|79.58|41.00|82.20|\||7.43|98.23|10.61|97.52|9.73|97.53|
> |I-EDL|47.41|86.29|45.38|83.52|42.36|82.29|\||6.95|98.33|27.40|95.29|9.57|97.58|
> |R-EDL|**37.60**|75.19|63.06|60.78|39.26|71.16|\||4.24|98.50|23.05|95.08|5.75|98.08|
> |Re-EDL|67.08|86.77|70.10|81.38|53.59|83.26|\||5.09|98.34|11.33|97.12|9.27|97.27|
> |ERNN|95.44|77.57|87.04|81.19|95.12|75.46|\||3.76|98.51|11.16|96.47|4.04|98.51|
> |D-EDL|73.34|85.48|63.19|**86.17**|43.71|84.85|\||4.86|98.79|**5.22**|98.41|8.53|97.98|
> |**Leaky-KL(ours)**|42.30|**86.28**|**44.08**|85.92|**35.59**|**87.89**|\||**2.01**|**99.56**|7.07|**98.43**|**2.90**|**99.29**|
>
> > Q3: There is limited discussion about how sensitive the performance is to this choice. For example, I cannot find the k values in Figure 4.
>
> We clarify the design of Figure 4. For a K-class task, k ranges from 0 to K, giving K+1 discrete points for leakage adjustment (solid lines), while coefficient tuning uses λ∈{0, 0.25, 0.5, 0.75, 1.0} (dashed lines). The x-axis represents normalized constraint strength (strong to weak): for coefficient tuning, this is λ itself; for top-k leakage adjustment, this is k/K. This normalization enables direct comparison of the two mechanisms across the same constraint spectrum. As shown, leakage adjustment demonstrates consistent and structured variation with k, and k=2 consistently emerges as optimal or near-optimal across all three medical datasets, demonstrating robustness to this choice. Moreover, leakage adjustment consistently matches or exceeds coefficient tuning across the entire constraint spectrum, confirming that the advantage is not specific to k=2 but holds broadly.
>
> > Q4: Including comparisons with non-EDL approaches would make the empirical claims stronger.
>
> We provide a comprehensive comparison with non-EDL baselines in our response to Reviewer 7nkc Q3. On KL-sensitive datasets (Endoscopy and Histopathology), Leaky-KL achieves significant improvements over all baselines (p<0.001), consistent with our central claim that anisotropic regularization addresses a structural limitation of isotropic KL constraint.
>
> > Q5: Although the intuition is fairly clear, the paper provides limited theoretical analysis explaining why anisotropic regularization should improve OOD detection.
>
> The key insight is that within the KL framework, direction is the only remaining degree of freedom once isotropic scaling is shown to be structurally limited. By allowing different classes to receive different effective constraint strengths, Leaky-KL mitigates the inherent coupling between ID vacuity preservation and OOD evidence suppression; a detailed justification is provided in our response to Reviewer 6sS8, Q1.

---

> > ### Author Rebuttal · Reviewer_se9V · 2026-04-03
> >
> > I appreciate the authors' responses to my questions. However, I am not fully convinced, particularly regarding the paper's novelty and the experimental evaluation (e.g., comparisons with state-of-the-art methods). Therefore, I would prefer to keep my rating unchanged.

---

> > > ### Author Response · Authors · 2026-04-03
> > >
> > > Dear Reviewer se9V,
> > >
> > > Thank you for your continued engagement. We would like to briefly recap our core contribution for clarity.
> > >
> > > The shift from isotropic to anisotropic regularization is not merely a mechanical modification, but a principled response to a *structural limitation* of the KL framework that cannot be resolved by coefficient tuning regardless of how well the coefficient is chosen. This is directly validated by consistent outperformance over the oracle isotropic baseline across all five datasets spanning both medical and natural image domains.
> > >
> > > We remain open to any further clarification should you have specific questions.

---

### Official Review · Reviewer_7nkc · 2026-03-12

**Soundness:** 1
**Presentation:** 2
**Significance:** 1
**Originality:** 1
**Overall Recommendation:** 2
**Confidence:** 4

**Summary:**

This paper identifies an "all-or-nothing dilemma": the KL divergence regularization on non-target classes either over-penalizes ID samples (when fully applied) or under-penalizes OOD samples (when removed). The proposed method, Leaky-KL, selectively removes the top-k predicted classes from KL regularization while constraining the rest. The article analyzes the key challenge of moving from isotropic (uniform coefficient scaling) to anisotropic (class-selective) regularization. Experiments are conducted on three medical imaging datasets for OOD detection. Overall, the main contribution is replacing uniform KL regularization scaling with a selective masking mechanism that allows high-evidence non-target classes to leak through unpenalized.

**Compliance With Llm Reviewing Policy:**

Affirmed.

**Final Justification:**

The rebuttal added useful experiments and clarifications, but my core concern remains: Leaky-KL is the best within the EDL family yet still underperforms simpler non-EDL baselines on multiple datasets. I keep my original assessment.

**Key Questions For Authors:**

1. How does Leaky-KL perform on non-medical OOD benchmarks (CIFAR-10/100 vs. SVHN, Textures, etc.)?
2. For OOD samples at test time, are the top-k classes meaningful or spurious?
3. How sensitive is the method to the evidence activation function g(·)?

**Limitations:**

Yes

**Strengths And Weaknesses:**

Strengths
- The method is simple: a top-k masking to the existing KL term.
- Gradient analysis is the strongest theoretical contribution. The derivation showing that coefficient tuning produces isotropic gradients, while Leaky-KL produces class-dependent effective coefficients through leak ratios is clean.
- Density distribution visualizations in Figure 5 suggest that Leaky-KL does lead to tighter ID peaks while maintaining OOD separation.
- The motivation that $\lambda_{KL}$ tuning exhibits a plateau effect, where performance changes dramatically at 0, is interesting. This suggests that the issue is structural rather than tuning $\lambda_{KL}$.

Weaknesses
- Limited experiments and evaluation. The paper evaluates on only 3 medical datasets with only 1 backbone (ResNet18). There are no experiments on standard OOD benchmarks (CIFAR-10/100, ImageNet) that would allow comparison with the broader literature and demonstrate generality beyond the medical domain. As a result, it's unclear whether anisotropic has advantages over isotropic regularization.
- It's unclear whether the "all-or-nothing dilemma" is a general phenomenon. The all-or-nothing framing relies on two datasets showing opposite preferences. With only 3 datasets, this could reflect some properties of the datasets rather than a fundamental limitation.
- Very limited empirical gains. In Table 1 results, Leaky EDL only marginally outperforms baselines and sometimes underperforms baselines. Some improvements are within standard deviation.
- Missing critical baselines. The paper doesn't compare against non-EDL OOD detection methods (e.g., energy-based methods, Mahalanobis distance, ViM, KNN-based approaches) that are commonly used in OOD detection.
- No theoretical justification for why exempting the top-k classes should improve OOD detection. For OOD samples, the top-k classes could contain spurious patterns that should be constrained rather than preserved.

---

> ### Author Rebuttal · Authors · 2026-03-29
>
> Dear Reviewer 7nkc,
>
> We sincerely appreciate the time and effort you have dedicated to reviewing our paper and providing us with valuable feedback. Here are our replies.
> > Q1: There are no experiments on standard OOD benchmarks.
>
> We have conducted additional experiments using CIFAR and MNIST as ID datasets. Leaky-KL achieves the best performance among EDL-based methods across all settings, confirming generalizability beyond the medical domain. Please refer to our response to Reviewer se9V, Q2 for detailed results.
> > Q2: It's unclear whether the "all-or-nothing dilemma" is a general phenomenon.
>
> We provide results across 5 datasets spanning medical/natural domains in the table below. As analyzed in Sec. 2.2, the opposing trends reflect intrinsic characteristics: high-separability datasets (MNIST, Endoscopy) prefer λ=0, while low-separability datasets (CIFAR10, Histopathology, Dermatology) prefer λ=1. This pattern holds consistently across both domains, confirming that the dilemma is a general phenomenon rather than a medical-domain artifact. More importantly, the plateau effect persists across all datasets within λ∈(0,1], confirming that the limitation of isotropic KL regularization is independent of dataset properties.
> |FPR95/AUROC|Endoscopy-OOD|Dermatology-OOD|Histopathology-OOD|CIFAR10-Texture|MNIST-Texture|
> |:-|:-:|:-:|:-:|:-:|:-:|
> |EDL(λ_KL=0)|50.82/91.12|92.27/68.23|94.60/78.84|78.09/84.13|10.68/96.75|
> |EDL(λ_KL=0.25)|61.14/89.97|87.08/70.37|79.84/81.80|50.60/82.21|55.30/89.35|
> |EDL(λ_KL=0.5)|59.51/90.16|87.33/70.21|79.38/81.94|46.49/82.87|55.76/88.98|
> |EDL(λ_KL=0.75)|59.91/89.94|87.15/69.35|79.52/81.86|48.72/82.27|55.33/89.53|
> |EDL(λ_KL=1.0)|59.68/90.12|87.16/70.30|79.67/81.77|46.27/82.15|53.79/90.78|
> |**Trend at λ_KL=0**|↓|↑|↑|↑|↓|
> |**Leaky-KL(ours)**|**35.95/92.32**|**88.40/71.94**|**74.63/83.94**|**44.08/85.92**|**7.07/98.43**|
> > Q3: Missing critical baselines on none-EDL methods and the empirical gains are limited.
>
> We provide a comparison including non-EDL baselines in the table below (paired t-test against Leaky-KL: * p<0.05, ** p<0.01, *** p<0.001). We further clarify the oracle nature of the optimal λ baseline in our response to Reviewer KPyK, Q1. Statistical significance on Endoscopy and Histopathology (p<0.001) confirms that gains are not marginal. The consistently weak performance across all EDL variants on Dermatology suggests inherent limitations of the EDL framework on this dataset, rather than a specific failure of Leaky-KL; within the EDL family, Leaky-KL maintains consistent improvement.
> ||Endoscopy||Dermatology||Histopathology||
> |-|-|-|-|-|-|-|
> ||**FPR@95**|**AUROC**|**FPR@95**|**AUROC**|**FPR@95**|**AUROC**|
> |MSP|55.92***|90.70**|84.06***|70.15*|74.02|82.01***|
> |MDS|67.43***|83.66***|**65.72**\***|71.04|66.49**|79.45***|
> |EBO|51.25**|91.03*|82.92**|71.71|74.92|82.88**|
> |ViM|49.53**|91.65|66.97***|**76.65**\**|61.00**|83.47|
> |KNN|57.95***|90.44**|86.43|66.19**|**59.62**\***|80.92***|
> |SCALE|50.53**|91.16*|82.92**|71.71|74.92|82.88**|
> |EDL(λ_KL=1)|59.68***|90.12**|87.16|70.30*|79.67*|81.77***|
> |EDL(λ_KL=0)|50.82***|91.12**|92.27*|68.23*|94.60***|78.84***|
> |EDL(optimal λ)|50.82***|91.12**|87.08|70.37|79.38|81.94***|
> |RED|64.65***|89.60**|85.58|70.94|86.46**|80.61***|
> |I-EDL|73.23***|88.33***|86.18|69.62*|74.35|82.18**|
> |R-EDL|61.37***|89.32**|85.79|70.28|77.30|82.19**|
> |Re-EDL|61.58***|89.72***|87.65|69.54|82.35**|79.90***|
> |ERNN|51.82***|90.91*|92.14**|68.25*|94.80***|78.62***|
> |D-EDL|48.71***|91.24*|79.67***|72.19|78.49*|82.16**|
> |**Leaky-KL(ours)**|**35.95**|**92.32**|88.40|71.94|74.63|**83.94**|
> > Q4: For OOD samples at test time, are the top-k classes meaningful or spurious?
>
> We clarify that top-k exemption is a training-time mechanism applied exclusively to ID samples; OOD samples are never seen during training. Their evidence patterns at test time are indirectly shaped by the KL constraint on training ID, encouraging low evidence on outliers without any test-time top-k selection.
> > Q5: No theoretical justification for why Leaky-KL should improve OOD detection.
>
> In the EDL framework, KL indirectly constrains OOD behavior through ID-only training, and its effects on ID/OOD vacuity are inherently coupled. Isotropic coefficient tuning amplifies this coupling uniformly, making it impossible to balance the two modes and forcing a binary switch instead. Leaky-KL mitigates this coupling by allowing different classes to receive different effective constraint strengths. Please refer to our response to Reviewer 6sS8, Q1 for detailed justifications.
> > Q6: How sensitive is the method to the evidence activation function g(·)?
>
> The choice of g(·) is a standard component of the EDL framework and is orthogonal to our contribution. We follow the same setting (exp) as existing EDL-based medical OOD detection methods (ERNN, D-EDL) for fair comparison, and exploring alternative activation functions remains a valuable direction for future work.

---

> > ### Author Rebuttal · Reviewer_7nkc · 2026-04-02
> >
> > I thank the authors for the additional experiments and responses.
> >
> > The CIFAR-10 and MNIST experiments and non-EDL baselines with statistical tests (Q3) strengthen the evaluation. The 5-dataset analysis makes the all-or-nothing dilemma more convincing. Thanks for the clarification that top-k masking is training-time only.
> >
> > I feel that the method still shows very limited empirical gains. On Dermatology, Leaky-KL underperforms multiple non-EDL baselines (ViM: 76.65 vs. 71.94 AUROC). On Histopathology, gains over the best non-EDL methods are mixed (Leaky-KL AUROC 83.94 vs. ViM 83.47, but ViM FPR@95 61.00 vs. 74.63). The method's clearest gains are on Endoscopy, but this is one dataset. On the new CIFAR-10 benchmark, Leaky-KL achieves 85.92 AUROC, which is competitive within EDL but unlikely to be competitive with SoTA non-EDL methods. The paper's claim is that anisotropic regularization fundamentally transcends isotropic approaches, but the empirical evidence shows a method that is the best within the EDL family but still comparable to or weaker than simpler non-EDL alternatives on multiple datasets.
> >
> > Therefore I maintain my score.

---

> > > ### Author Response · Authors · 2026-04-03
> > >
> > > Dear Reviewer 7nkc,
> > >
> > > Thank you for your thoughtful follow-up and for acknowledging the strengthened evaluation. We appreciate the observation regarding non-EDL baselines—it points to a genuinely exciting direction.
> > >
> > > We agree that feature-space methods achieve strong performance through complementary mechanisms that are independent of evidential regularization. The question of *how to regularize evidence within EDL* is a distinct and open problem, and Leaky-KL addresses it directly. Across all three medical datasets as well as the natural image benchmarks, Leaky-KL consistently outperforms all EDL variants including the oracle isotropic tuning baseline, validating that anisotropic regularization accesses a solution space structurally unreachable by isotropic scaling. We see the strong performance of feature-space methods not as a limitation, but as motivation for future work combining anisotropic regularization with feature-aware scoring.
> > >
> > > We thank the reviewer again for the constructive engagement throughout the review process.

---

### Official Review · Reviewer_6sS8 · 2026-03-12

**Soundness:** 2
**Presentation:** 2
**Significance:** 2
**Originality:** 2
**Overall Recommendation:** 4
**Confidence:** 2

**Summary:**

This paper revisits the role of KL regularization in evidential deep learning for medical OOD detection. Rather than taking the standard positions of full regularization or no regularization, the paper argues that both are too crude and proposes Leaky-KL, which selectively allows evidence leakage for top-k predicted classes while constraining the rest. The central claim is that anisotropic regularization is fundamentally better than isotropic coefficient tuning.

**Compliance With Llm Reviewing Policy:**

Affirmed.

**Key Questions For Authors:**

N/A

**Strengths And Weaknesses:**

## Strengths

- The paper asks a focused and important question in a safety-critical setting.
- The "all-or-nothing dilemma" is a clear framing and gives the paper a precise conceptual target.
- The proposed intervention is simple enough to be legible while still being meaningfully different from standard coefficient tuning.
- The reported consistency across three medical datasets strengthens the practical relevance of the idea.

## Weaknesses

- The paper's strongest claim is not that Leaky-KL works, but that the old framing is wrong. That is a bigger claim, and the visible evidence needs to carry more of that weight. Beating fixed coefficients on several datasets is encouraging, but it does not yet prove that anisotropy is the right principle rather than just a better local heuristic.
- The selective top-k leakage rule may be effective, but right now it still looks like a hand-designed control knob attached to a criticism of hand-designed control knobs. The paper argues against crude scalar tuning, then introduces a more structured form of tuning. That may still be progress, but the paper should be more honest about how much theory versus engineering is actually doing the work.
- The paper's framing suggests that existing EDL approaches fail because they constrain uncertainty in the wrong shape. If that is true, then the most important evidence is not just aggregate improvement, but a convincing demonstration that Leaky-KL changes the uncertainty geometry in the way the paper claims. Otherwise the method risks being remembered as "the top-k variant that happened to benchmark well."

---

> ### Author Rebuttal · Authors · 2026-03-29
>
> Dear Reviewer 6sS8,
>
> We sincerely appreciate the time and effort you have dedicated to reviewing our paper and providing us with valuable feedback. Here are our replies.
>
> > Q1: The claim that anisotropy is the right principle rather than a better local heuristic.
>
> We appreciate this question and would like to clarify that our central claim is that isotropic tuning is *structurally limited*, rather than fundamentally wrong. KL is central to EDL's OOD detection capability as the core mechanism for suppressing non-target evidence. However, ID-only training inherently couples the effects of KL on ID vacuity preservation and OOD evidence suppression. As shown in the dashed lines of Fig. 4, once λ>0, performance fluctuates within a narrow range without meaningful improvement, confirming that this coupling cannot be resolved by λ tuning and that the limitation is structural. Within the KL framework, the only remaining degree of freedom is therefore *direction* (which classes to constrain) rather than *scaling* (how much to constrain), making anisotropic regularization a principled direction rather than a heuristic choice.
>
> In practice, Leaky-KL as an anisotropic method applies asymmetric constraint on ID and OOD samples: for ambiguous ID, top-k exemption mitigates over-penalization by preserving evidence for genuinely similar classes; for OOD, whose evidence patterns do not align with any ID class, the remaining constraint continues to suppress spurious evidence accumulation. As shown in Fig. 5, compared to λ=1, Leaky-KL produces tighter ID distributions with less over-penalization; compared to λ=0, OOD distributions are less overconfident with higher uncertainty, confirming that anisotropic regularization simultaneously addresses both failure modes of the isotropic extremes, therefore improving OOD detection.
>
> > Q2: The selective top-k leakage rule still looks like a hand-designed control knob attached to a criticism of hand-designed control knobs.
>
> We find this framing insightful: the knob analogy in fact helps sharpen our argument. Our criticism is not of hand-designed mechanisms per se, but of λ as a meaningful knob — which it is not. As shown in Fig. 4, the plateau effect reveals that λ tuning behaves more like a **switch**: the critical transition occurs at λ=0, after which performance remains largely flat regardless of the coefficient value. It is not a knob with a useful range, but a binary choice between full constraint and no constraint.
>
> We do not deny that top-k involves design choices; any concrete implementation of anisotropic regularization will. Our contribution is not the specific mechanism of top-k, but rather the identification of the structural limitation of isotropic scaling and the establishment of anisotropic direction as the necessary path forward within the KL framework. Leaky-KL serves as a simple and effective instantiation of this direction; whether top-k is the optimal implementation remains an open question that we explicitly acknowledge as future work.
>
> > Q3: The most important evidence should be a convincing demonstration that Leaky-KL changes the uncertainty geometry in the way the paper claims.
>
> We appreciate this request for stronger geometric evidence. In our framework, the relevant geometry is that of the regularization constraint rather than uncertainty directly. Since OOD samples are absent during training, KL regularization shapes uncertainty indirectly by constraining the evidence space on ID training samples. As illustrated in Fig. 3, Classic KL with varying λ produces gradients that scale along a fixed direction, achieving only one-dimensional isotropic adaptation. Leaky-KL, by contrast, allows different classes to have different leak ratios, producing different gradient directions, spanning a two-dimensional parallelogram region. This expanded optimization space enables the model to better adapt to data distribution patterns.
>
> Further empirical support comes from leak ratio analysis on the Bone Marrow dataset (9 classes), with full results in our response to Reviewer KPyK, Q4. The train/val agreement (Spearman ρ=0.98) confirms that leak ratios reflect intrinsic data properties stable across splits. Moreover, the class-dependent variation is substantial: EOS (eosinophils, morphologically distinctive with characteristic bilobed nucleus and granule-laden cytoplasm) receives a leak ratio of 0.0028, while LYT (lymphocytes, exhibiting high morphological variability) receives 0.3207. This directly confirms that anisotropic regularization adapts to data-intrinsic class properties rather than arbitrary design choices: classes with distinctive morphology receive stronger constraint, while visually ambiguous classes are selectively exempted, achieving the directional adjustment in gradient space illustrated in Fig. 3.

---

> > ### Author Rebuttal · Reviewer_6sS8 · 2026-04-05
> >
> > N/A

---

> > > ### Author Response · Authors · 2026-04-06
> > >
> > > Dear Reviewer 6sS8,
> > >
> > > Thank you for your thorough engagement and for acknowledging that your concerns have been fully addressed. We are glad that our responses were helpful.

---

### Decision · Program_Chairs · 2026-04-30

**Decision:**

Reject

**Comment:**

This paper studies KL regularization in Evidential Deep Learning for medical OOD detection and introduces Leaky‑KL, an anisotropic regularization scheme that selectively relaxes constraints on top‑predicted classes. Reviewers agreed that the paper addresses an important question and appreciated the clear framing of the "all‑or‑nothing dilemma", the simplicity of the proposed modification, and the authors’ thorough engagement during rebuttal, including additional experiments and analyses. The work is technically coherent and shows consistent improvements over prior EDL variants, demonstrating promise within the EDL family.

However, reviewers raised concerns about the paper’s significance for ICML -- the concerns remained pretty much unchanged during the rebuttal, even with the strong engagement of the authors. Despite the strengthened evaluation, the empirical gains are modest and confined to comparisons within the EDL family, with performance that is competitive with (but not superior to) simpler non‑EDL baselines on several datasets. Conceptually, the contribution is viewed as an incremental but well‑motivated design refinement rather than a broadly impactful advance in uncertainty modeling or OOD detection. The key claims about anisotropic regularization representing a superior principle are not fully supported by the limited evaluation (single backbone, selective benchmarks). For these reasons, while the paper is promising, the reviewers concluded that it still needs some work before being accepted at ICML.